# Insights into the Molecular Basis of Huanglongbing Tolerance in Persian Lime (*Citrus latifolia* Tan.) through a Transcriptomic Approach

**DOI:** 10.3390/ijms24087497

**Published:** 2023-04-19

**Authors:** Humberto Estrella-Maldonado, Carlos González-Cruz, Cristian Matilde-Hernández, Jacel Adame-García, Jorge M. Santamaría, Ricardo Santillán-Mendoza, Felipe Roberto Flores-de la Rosa

**Affiliations:** 1Instituto Nacional de Investigaciones Forestales, Agrícolas y Pecuarias (INIFAP), Campo Experimental Ixtacuaco, Km 4.5 Carretera Martínez de la Torre-Tlapacoyan, Cong. Javier Rojo Gómez, Tlapacoyan C.P. 93600, Veracruz, Mexico; 2Tecnológico Nacional de México, Campus Úrsulo Galván, Km 4.5 Carretera Cd. Cardel-Chachalacas, Úrsulo Galván C.P. 91667, Veracruz, Mexico; 3Centro de Investigación Científica de Yucatán A.C., Calle 43 No. 130, Colonia Chuburná de Hidalgo, Mérida C.P. 97205, Yucatán, Mexico

**Keywords:** basal resistance, citrus, Huanglongbing, transcriptome, pathogen triggered immunity

## Abstract

Huanglongbing (HLB) is a vascular disease of *Citrus* caused by three species of the α-proteobacteria “*Candidatus* Liberibacter”, with “*Candidatus* Liberibacter asiaticus” (*C*Las) being the most widespread and the one causing significant economic losses in citrus-producing regions worldwide. However, Persian lime (*Citrus latifolia* Tanaka) has shown tolerance to the disease. To understand the molecular mechanisms of this tolerance, transcriptomic analysis of HLB was performed using asymptomatic and symptomatic leaves. RNA-Seq analysis revealed 652 differentially expressed genes (DEGs) in response to *C*Las infection, of which 457 were upregulated and 195 were downregulated. KEGG analysis revealed that after *C*Las infection, some DEGs were present in the plant–pathogen interaction and in the starch and sucrose metabolism pathways. DEGs present in the plant–pathogen interaction pathway suggests that tolerance against HLB in Persian lime could be mediated, at least partly, by the *ClRSP2* and *ClHSP90* genes. Previous reports documented that *RSP2* and *HSP90* showed low expression in susceptible citrus genotypes. Regarding the starch and sucrose metabolism pathways, some genes were identified as being related to the imbalance of starch accumulation. On the other hand, eight biotic stress-related genes were selected for further RT-qPCR analysis to validate our results. RT-qPCR results confirmed that symptomatic HLB leaves had high relative expression levels of the *ClPR1*, *ClNFP*, *ClDR27*, and *ClSRK* genes, whereas the *ClHSL1*, *ClRPP13*, *ClPDR1*, and *ClNAC* genes were expressed at lower levels than those from HLB asymptomatic leaves. Taken together, the present transcriptomic analysis contributes to the understanding of the *C*Las-Persian lime interaction in its natural environment and may set the basis for developing strategies for the integrated management of this important *Citrus* disease through the identification of blanks for genetic improvement.

## 1. Introduction

Citrus cultivation is an economically important global activity, resulting in concentrates, food supplements, beverages, as well as essential oils and citric acid [1]. Globally, Mexico has become the fifth largest producer of citrus; moreover, Mexico is the second largest exporter of limes and lemons in the world, sending 18% of its sales to the United States and Japan [2]. Currently, the country produces a total of 6,851,000 tons of citrus, of which 27.4% correspond to lemons and limes. Within Mexico, Veracruz State is the main producer of these crops [3], and moreover, Veracruz State has positioned itself as the most important producer and exporter of Persian lime (*Citrus latifolia* Tan.) in the world [4].

Citrus crops are susceptible to diseases caused by bacteria, viruses, fungi, mycoplasmas, and other similar organisms [5]. Consequently, at the end of 2014, Huanglongbing (HLB) disease was reported in 16 of Mexico’s 23 citrus-producing states [6]. HLB, a disease caused by *Candidatus* Liberibacter asiaticus (*C*Las), is considered a serious threat to world citrus production [7], even more destructive than other vascular diseases. Treatments for *C*las-infected trees are still experimental and not yet available to producers [8]. Thus, it is important to understand the molecular mechanisms of HLB-tolerant genotypes as a first step for genetic improvement of these economically important species [9].

For instance, *C*Las is a Gram-negative bacterium resident in the phloem of host plants that causes systemic diseases in citrus [10]. Furthermore, the Asian Citrus Psyllid (*Diaphorina citri* Kuwayama) is the transmitting vector of the bacterium through feeding in the new flushes and grafting infected buds [7], reducing the reproductive physiology of young and adult plants of all species and hybrids of the *Citrus* genus [11,12,13]. Symptoms of HLB include spotting, leaf starch accumulation, stunted plant growth, zinc deficiency, fruit deformation, poor juice quality, and reduced root biomass [14]. Likewise, HLB causes nutrient imbalances that worsen the physiological and functional status of infected plants [15]. Thus, the cellular integrity of the plant is compromised due to reactive oxygen species (ROS) [16]. *C*Las-infected Persian lime trees show an increase in starch contents in the mesophyll, while in *C*Las-infected sweet orange trees it concentrates in the palisade parenchyma; however, in both cases, starch accumulation results in hyperplasia, which causes a collapse in the phloem [17].

Although physiological and histological symptoms have been observed in most citrus trees diseased with HLB, causing important economic losses for citriculture throughout the world [17,18], some citrus genotypes have been reported as being tolerant to the *C*Las infection [19]. Among them, Persian lime has been reported as being the least susceptible species to this disease [20]. Recent evidence suggests that tolerance to HLB is associated with better activity in phloem dynamics compared with that of Mexican limes and oranges [21], but the molecular mechanisms of this tolerance remain unclear.

To understand the plant response to the *C*Las infection, multiple transcriptomic analyses of gene expression profiles by RNA sequencing (RNA-Seq) have been performed. Studies have been implemented on highly susceptible citrus cultivars and genotypes [15,22], but few studies have investigated HLB-tolerant citrus genotypes [19,23]. One such study showed that in asymptomatic HLB leaves from the HLB-tolerant Mexican lime (*C. aurantifolia*), the expression of genes belonging to isoprenoid secondary metabolism, hormone signaling pathways, and phosphoinositide signaling increased. On the contrary, in HLB-symptomatic leaves, the expression of genes belonging to light reactions, phenolic secondary metabolism, redox reactions, and ubiquitin degradation increased [23]. Furthermore, these authors reported that few genes related to carbohydrate secondary metabolism were upregulated in symptomatic-HLB leaves and, therefore, no significant changes were observed in carbohydrate metabolism in Mexican lime. These results are very important to elucidate the molecular mechanism of HLB tolerance in Mexican lime.

In the present study, we analyze the transcriptomic differences between asymptomatic and symptomatic Persian lime leaves in response to natural infection with *C*Las. This is the first report of the transcriptomic profile of the tolerant Persian lime in response to HLB diseases globally. The aim is to identify differentially expressed genes induced by HLB natural infection that may be associated with tolerance to the disease and to understand the molecular basis of such tolerance.

## 2. Results

### 2.1. Visual Symptoms and Confirmation of the Presence of CLas by RT-qPCR Analysis

HLB symptoms observed in Persian lime (*Citrus latifolia* Tan.) trees grown at Campo Experimental Ixtacuaco in the northern region of Veracruz, Mexico, were consistent with those reported in other *C*Las-infected citrus trees (Figure 1). Leaf samples were collected at different points within the experimental orchard, based on the presence of different HLB symptoms (such as leaf chlorosis and starch accumulation in veins). RT-qPCR analysis confirmed that symptoms were caused by the presence of *C*Las; likewise, plants that did not show HLB symptoms and had a quantification cycle (Cq) ≥ 38 were considered negative for HLB detection analysis (Figure 1).

### 2.2. Results from the Assembly of the Transcriptome from Persian Lime

Transcriptome assembly using the Trinity program showed a total coverage of 118,955,352 base pairs, 247,494 transcripts, a GC percentage of 42%, and an N50 of 1532. Likewise, 38,638 contigs appeared with a length of 1000 bp. Results from the transcriptome assembly from *C*Las-infected or uninfected Persian lime showed high-quality values Appendix A.

### 2.3. Identification of Differentially Expressed Genes between HLB-Asymptomatic and HLB-Symptomatic Persian Lime Plants

Using RNA-Seq technology, it was possible to identify differentially expressed genes (DEGs) in both conditions (Figure 2). Thus, with the filter criterion of *p*-adjusted < 0.05, 652 DEGs were detected when comparing HLB-asymptomatic with HLB-symptomatic Persian lime plants Appendix A. The heat map plot of these 652 DEGs showed a different expression pattern between HLB-asymptomatic and HLB-symptomatic leaves (Figure 2a). Thus, 195 DEGs were downregulated in HLB-symptomatic leaves, while 457 DEGs were upregulated. The Venn diagram shows the differences and overlapping of accumulated transcripts between HLB-asymptomatic and HLB-symptomatic-HLB conditions (Figure 2b). Interestingly, the analysis also revealed that 169 genes (25.9%) were expressed in both conditions. In contrast, 112 genes (17.1%) were expressed only in HLB-asymptomatic leaves, while 371 genes (56.9%) were expressed only in HLB-symptomatic leaves. In this case, the heat map (Figure 2a) and Venn diagram (Figure 2b) analyses showed strong evidence that HLB-symptomatic plants had a large number of upregulated DEGs induced by HLB disease.

### 2.4. Functional Classification of DEGs Found in HLB-Symptomatic Leaves

Gene ontology (GO) analysis revealed three major categories (biological process, molecular function, and cellular component) and 43 subcategories for up- and downregulated genes. The HLB-symptomatic leaves showed a greater number of upregulated than downregulated genes in the Biological Process category, which includes cellular component organization or biogenesis process, primary metabolic process, response to abiotic stimuli, developmental process, and regulation of biological processes—the subcategories where the highest number of upregulated genes (green bars) were observed (Figure 3).

Likewise, within the Molecular Function category, catalytic activity, hydroxylase activity, ion binding, oxidoreductase activity, and binding were the subcategories with the highest number of upregulated genes (green bars) occurring in symptomatic leaves; however, nucleic acid transcription factor activity was the only subcategory that showed a greater number of downregulated genes (red bars). 

Concerning the Cellular Component category, extracellular, cell wall, and plasmodesma subcategories were excluded; HLB-symptomatic leaves showed a higher number of upregulated genes than those downregulated (Figure 3). Interestingly, up-regulation (green bars) of a large number of chloroplast- and membrane-related genes was observed in diseased leaves.

### 2.5. Gene Pathway Enrichment Analysis of Host Pathways in Response to CLas Infection

The results obtained from the KEGG analysis showed that major plant metabolic pathways were affected by *C*Las infection on symptomatic leaves (Appendix A). Two pathways that were significantly affected by HLB were the plant–pathogen interaction pathway (Figure 4; Appendix A) and the starch and sucrose metabolism pathway (Figure 5; Appendix A). 

In the plant–pathogen interaction pathway, 4 genes were downregulated (red box) in response to HLB: *Cf-9* (disease resistance protein), *FLS2* (LRR receptor-like threonine-protein kinase), *XA21* (receptor kinase-like protein), and *RPM1* (disease resistance protein). On the contrary, 3 genes were upregulated (green box): *PR1* (pathogenicity-related protein 1), *RPS2* (disease resistance protein), and *HSP90* (heat shock protein 90KDa beta) (Figure 4). 

On the other hand, in the starch and sucrose metabolic pathways, 9 genes were upregulated (green box) by the *C*Las infection: *MGAM* (maltase-glucoamylase; EC:3.2.1.20 3.2.1.3), *GPI* (glucose-6-phosphate isomerase; EC:5.3.1.9), *GCK* (glucokinase; EC:2.7.1.2), *glgC* (glucose-1-phosphate adenylyl transferase; EC:2.7.7.27), *glgA* (starch synthase; EC:2.4.1.21), beta-amylase (EC:3.2.1.2), *glgP* (glycogen phosphorylase; EC:2.4.1.1), *WAXY* (granule-bound starch synthase; EC:2.4.1.242) and *GBE1* (1,4-alpha-glucan branching enzyme; EC:2.4.1.18) (Figure 5). 

### 2.6. RT-qPCR Validation of DEGs from RNA-Seq Analysis

We identified and selected DEGs involved in biotic stress that showed up-regulation (log2 FC > 2) and high abundance counts, and downregulation (log2 FC < 2) and low or no abundance counts, to validate by RT-qPCR analysis the RNA-Seq data (Table 1; Appendix A). Thus, eight DEGs were selected: 4 upregulated genes: *ClPR1* (DN9557_c0_g1_i1), *ClNFP* (DN91145_c0_g1_i1), *ClDRL27* (DN14097_c0_g1_i1), and *ClSRK* (DN6810_c0_g2_i1); and 4 downregulated genes in HLB-symptomatic leaves: *ClHSL1* (DN112451_c0_g1_i1), *ClRPP13* (DN27097_c0_g1_i1), *ClPDR1* (DN14613_c0_g1_i1), *ClNAC* (DN10036_c0_g2_i1) (Table 1). 

The RT-qPCR analysis confirmed that upregulated genes in HLB-symptomatic leaves also showed higher relative expression levels than asymptomatic ones (Figure 6a). Likewise, RT-qPCR analysis also confirmed that those genes that were downregulated in HLB-symptomatic leaves in the transcriptome data also showed low relative expression levels (Figure 6b). Thus, the results obtained for these sets of up- and down-regulated genes in HLB-symptomatic plants, agreed with the data from the transcriptome, thus validating the observed differential gene expression.

## 3. Discussion

HLB is economically the most important disease in the citrus industry globally. Recently, HLB was described as a pathogen-triggered immune disease or autoimmune disease, as the main damage caused in infected plants results from an uncontrolled immune response such as callose deposition and overproduction of H_2_O_2_, rather than from a direct *C*Las activity [24]. This could be especially relevant to understand the molecular mechanisms of highly susceptible and tolerant genotypes and to develop alternative management tools [9,25]. Persian lime (*Citrus latifolia* Tan.) is an important and underestimated model for studying the molecular response of *Citrus* to HLB, especially due to the low deposition of starch, callose, and proteins in the phloem of HLB-infected trees [17]. This is in line with the fact that HLB-infected Persian lime trees are able to maintain physiological and biochemical detoxification activities [21], which allow them to maintain high yields despite being infected. 

The present work studied transcriptional responses in naturally infected Persian lime trees with symptoms corresponding to *C*Las infection. The differences in transcript per million and fold change of related biotic stress DEGs in response to HLB showed interesting evidence to understand the differences between HLB asymptomatic and symptomatic Persian lime leaves. For instance, the *ClPR1*, *ClNFP*, *ClDRL27*, and *ClSPK* genes were upregulated in symptomatic HLB Persian lime leaves, likewise, the high expression levels of these genes confirmed by RT-qPCR analysis, validated the RNA-Seq study. 

Therefore, it is confirmed that these genes could play an important role in the response of Persian lime to the biotic stress caused by HLB. Concerning the *ClHLS1*, *ClRPP13*, *ClPDR1*, and *ClNAC* genes, the low counts of transcripts per million and low expression levels validated by both RNA-Seq and RT-qPCR showed that these genes were repressed when the Persian lime leaves were infected with *C*Las. Although several studies have reported DEGs in HLB-susceptible and HLB-tolerant citrus genotypes [19,26,27], the identification and expression of DEGs in Persian lime, a species considered HLB-tolerant, had not been studied. 

The identification of the *ClPR1* gene in the transcriptome of HLB-symptomatic Persian lime leaves is relevant because this gene plays a key role in multiple signaling pathways in plant immunity [28]. The role of this gene in the response to HLB is not clear yet, but its high expression has been associated with the enhancement of disease tolerance in transgenic lines of *C. sinensis* [27]. In addition, it has been reported that *C*Las infection induces the expression of *PR* genes, including *PR1*, which showed an expression peak at 21 days after bud initiation on infected *C. sinensis* trees [24]. Nevertheless, it has also been observed that the *C*Las effector *SDE15* can strongly suppress *PR1* expression in *C. paradisi* [29]. Further studies are indeed necessary to elucidate if the activity of *ClPR1* is associated with HLB-tolerance in Persian lime or if it is an effect of the manipulation of the plant immune system by CLas.

The present study is the first transcriptomic analysis to reveal the molecular mechanisms of the HLB tolerance exhibited by Persian lime. Other transcriptomic studies in highly susceptible *Citrus* species such as *C. sinensis* show a large number of genes downregulated in response to *C*Las infection, especially transporters, transcription factors, and signaling receptors [30,31]. In contrast, RNA-Seq analysis of *P. trifoliata* and hybrids suggests that there are a large number of genes and metabolic pathways upregulated in response to HLB in tolerant genotypes [32]. Our transcriptomic analysis in Persian lime demonstrated that, in response to HLB, a great number of genes were upregulated, similar to what was observed in Mexican lime [23].

A KEGG analysis of two very important metabolic pathways sheds light on the mechanisms associated with the tolerance towards HLB shown by Persian lime. Within the plant–pathogen interaction pathway (Figure 5), the *Cf-9* gene is downregulated; this gene is a calcium-dependent rapidly elicited protein (ACRE) related to HR response [33]. This result is similar to that reported in *C. sinensis*, where *Cf-9* was repressed by *C*Las infection [34]. Additionally, the *FLS2* gene was also downregulated in the transcriptome; this gene has been identified as a kinase receptor that activates PTI downstream by the flagellin perception [35]. The expression of this gene has been reported as a PTI marker using the flg22 peptide from *Xanthomonas citri* var. *campestri* (*Xccflg22*) and *C*Las (*CLasflg22*); however, the *FLS2* expression is lower when induced by *CLasflg22* than by *Xccflg22* in HLB-susceptible genotypes [36]. Other transcriptomic studies have shown that *FLS2* is downregulated or absent in leaves of susceptible citrus infected with *C*Las [37,38]. Previously, it was reported that the *C*Las movement *in planta* is not through flagella in susceptible citrus, which allow *C*Las to avoid PTI activation [39]. Thus, our results suggest that the tolerance to HLB in Persian lime is not through sensing flagellin protein from *C*Las; however, the infection alters the activity of its receptor, and further studies are necessary to elucidate the role of *FLS2* in the establishment of HLB. 

In addition, the *Xa21* gene was also downregulated in symptomatic Persian lime leaves; this gene was first reported in *Oryza longistaminata* as a resistance gene against bacterial infections [40], and it is used in the genetic improvement of *C. lemon* [41] and W. Murcott mandarin (*Citrus* sp.) [42] against bacterial canker, but there are no reports of this gene activity in resistance or tolerance against HLB. However, it is a good candidate for the *C*Las response, as *Xa21* recognizes a tyrosine-sulfated protein from gram-negative bacteria [43], and *C*Las secretes many proteins considered nonclasically secreted proteins (ncSecPs) with sulfate groups [44], so the repression of this gene may be a mechanism used by *C*Las to alter PTI activation.

Interestingly, in our results, the *ClRPM1* gene is repressed in the HLB symptomatic leaves, while the *ClRIN4* gene is not altered in their activity (Figure 4). The *RPM1* gene has been reported as being upregulated in susceptible citrus genotypes [37], which could be associated with the upregulated expression of the *RIN4* gene [45] as these genes have shown to have a positive regulatory interaction [46]. Recently, it was demonstrated that the overexpression of *RIN4* aids in the establishment of the *C*Las infection and the development of HLB symptoms [47]. 

Our results revealed a possible critical molecular mechanism of HLB tolerance in Persian lime: the activation of ETI through the *ClRPS2* activity and preventing the severity of *C*Las infection by not altering the *ClRIN4* expression, which has been previously associated with tolerance against HLB in hybrids [32]. Clearly, further studies are necessary to evaluate this hypothesis. The gene *ClRPS2* was upregulated in the symptomatic *C. latifolia* transcriptome, which has been observed to have a negative regulation by the *RIN4* gene [48]. The *RPS2* gene is considered a nucleotide-binding domain leucine-rich repeat (NLR) receptor that is associated with the triggering of immunity against a broad range of plant pathogens [49]. 

Additionally, in the plant–pathogen pathway, other two genes were upregulated by *C*Las infection: *PR1*, a gene related directly to the PTI [50] that is repressed directly by a pathogenicity effector of the *C*Las [29], as discussed previously, and the *HSP90* gene, which has been associated with an elicited defense response to the *C*Las [51]. HSP90 is a heat shock protein (HSP) widely associated with various signaling proteins related to disease resistance in plants [52]. HSP90 functions as a molecular chaperone, associated with the activation of salicylic acid biosynthesis [53] and other hormones like auxins and jasmonic acid [54]. Metabolomics studies have shown that tolerance against HLB is strongly controlled by phytohormones such as auxins and cytokinins, which are responsible for plant growth and phloem regeneration [55], which physiological features are observed in Persian lime during *C*Las infection [21]. Thus, our results suggest that another possible mechanism of HLB tolerance in Persian lime is via the hormonal response mediated by *ClHSP90*, which could be in turn mediated by the upregulation of the *ClRPS2* gene (Figure 4).

The second pathway involves the starch and sucrose metabolism pathways. Starch accumulation is one of the most important physiological traits observed in the HLB-diseased trees [56]. In fact, it has been reported that *C*Las can produce effectors with the ability to induce an overaccumulation of this carbohydrate [57]. Therefore, it was relevant to identify in the present study nine genes belonging to the starch and sucrose metabolism pathways that were upregulated in the HLB-symptomatic leaves. First, the *MGAM* (maltase-glucoamylase) gene was identified in the Persian lime transcriptome. Although very little information exists for this gene in plants, evidence suggests that it is involved in starch degradation [58], participating in the hydrolytic starch degradation route [59]. In addition, the *GPI* (glucose-6-phosphate isomerase) gene was also upregulated; this gene is related to both catabolic glycolysis and anabolic gluconeogenesis [60], and its activity has been associated with tolerance against abiotic stress [61]. It is important to note that in *C. sinensis*, the downregulation of this gene has been associated with the puffing disorder of the fruits [62]. Thus, our results suggest that the upregulation of the *ClGPI* gene could be associated with the tolerance of Persian lime against HLB through maintenance of the primary metabolism, contrary to what occurs in susceptible genotypes [63]. A *GCK*-like gene was also upregulated; this kind of hexokinase has been related to the phosphorylation of glucose to obtain D-glucose-6P [64], which could be associated with starch accumulation [63]. Additionally, it was observed in the transcriptome that genes associated with starch biosynthesis were upregulated, such as *glgC*, *glgA*, and *glgP* (Figure 5), similar to those reported in susceptible citrus genotypes [65,66] and the *WAXY* gene associated with the formation of starch granules [67], as well as the *GBE1* gene related to the transformation of amylose into starch [68]. All of this is in line with the high levels of starch detected previously by our research team [69]. 

It is also important to note that a member of the beta-amylase family (*E3.2.1.2*) is upregulated in symptomatic leaves; this gene family is known for its ability to hydrolyze starch to maltose [70] and its downregulation has been reported in CLas-infected *C. sinensis*, while its expression is not affected in tolerant genotypes [65], thus suggesting that the Persian lime response to *C*Las infection may represent an attempt to maintain equilibrium between starch synthesis and metabolism. 

## 4. Materials and Methods 

### 4.1. Plant Material and Experimental Design

Persian lime (*Citrus latifolia*) leaf samples were obtained from an experimental orchard established at the Campo Experimental Ixtacuaco belonging to the Instituto Nacional de Investigaciones Forestales, Agrícolas y Pecuarias (INIFAP), located at the coordinates 20°2′35.48″ N and 97°5′52.60″ W. Persian lime trees grafted on Citrumelo Swingle rootstock were 5 years old at the time of sampling. A total of three visually healthy and physiologically mature (V6 according to the [71] scale) leaves from three asymptomatic trees (absence of visible symptoms of HLB) and three leaves from three different HLB-symptomatic trees (spotted leaves, starch accumulation in veins, etc.) were collected (Figure 1). All sampled leaves were immediately frozen in liquid nitrogen. RT-PCR confirmed the absence of other vascular pathogens (CTV and phytoplasma).

### 4.2. DNA Extraction and CLas Detection Persian Lime Leaves

A total of 200 mg of tissue were macerated in liquid nitrogen with a mortar and pestle for DNA extraction using the Plant DNA Purification Kit (Norgen BIOTEK Corp., Thorold, ON, Canada), according to the manufacturer’s instructions. Detection of CLas was achieved by RT-qPCR assay using primers HLB-4G (5′AGTCGAGCGCGTATGCGAAT-3′) and HLBr (5′-GCGTTATCCCGTAGAAAAAGGTAG-3′) following the protocol reported by [72]. Amplifications were performed using a thermocycler CFX-96 Real-Time PCR System in a 96-well PCR plate (Bio-Rad, Hercules, CA, USA) and SsoAdvanced Universal Inhibitor-Tolerant SYBR^®^ Green Supermix (Bio-Rad) for signal detection. All reactions were performed in triplicate in 10 µL reaction volumes, using 200 ng of DNA per reaction. Plants were considered PCR-positive for CLas when the CT (cycle threshold) value was below 34.

### 4.3. Library Construction and Sequencing

RNA from leaf tissue was processed using Direct-zol™ RNA MiniPrep (Zymo Research). The RNA pellet was re-suspended in 30 µL RNase-free water. RNA concentration and purity were measured using a NanoDrop One^®^ spectrophotometer (Thermo Scientific NanoDrop Technologies, LLC, Wilmington, DE, USA). The RNA integrity was evaluated by 1.5% agarose gel electrophoresis during 30 min at 80 V. RNA-Seq was constructed using the NextSeq 500 Illumina platform, following the TruSeq^TM^ Stranded RNA Library Prep Kit (Illumina^®^, San Diego, CA, USA) according to the manufacturer’s instructions. The quality of the six cDNA libraries was tested using an Agilent 2100 Bioanalyzer (Agilent, Santa Clara, CA, USA). Library preparation and sequencing of cDNA libraries were carried out at the University Unit of Massive Sequencing and Bioinformatics of the Institute of Biotechnology of the National Autonomous University of Mexico (UUSMB IBT-UNAM). A total of six cDNA libraries were sequenced: three asymptomatic HLB leaf libraries and three symptomatic HLB leaf libraries. After sequencing, the quality control of raw RNA-Seq reads was performed using FastQC (version 0.11.3) to remove low-quality reads (Q > 30) (http://www.bioinformatics.babraham.ac.uk/projects/fastqc/ accessed on 1 March 2022). Reads above 32 nt, without the presence of adapters, were considered for further analysis. Alignments of filtered reads were performed with Smalt software (version 0.7.6) (https://bioweb.pasteur.fr/packages/pack@smalt@0.7.6 accessed on 7 March 2022) using *Arabidopsis thaliana* and *C. sinensis* as reference genomes. Subsequently, these filtered reads were used for *de novo* assembly of the Persian lime transcriptome using Trinity software (version 2.4) [73] with default parameters. For assembly quality analysis, metrics such as the total number of contigs, longest contig length, average and median contig length, L50, and N50 were calculated using Quast v. 5.2.0 software [74].

### 4.4. Differential Expression Analysis and Transcriptome Annotation

Differential expression analysis was performed between asymptomatic and HLB-symptomatic Persian lime leaves. Reads were mapped to be de novo assembled using Bowtie2 v. 2.4.5 software [75]. Transcript abundance was calculated using the Trinity bundle with RSEM v. 1.3.3 and Bowtie2 v. 2.4.5 software [76]. The IDEAMEX website [77] is a platform for differential expression analysis of RSEM data based on the DESeq2 method [78]. Tests for significance with the Poisson distribution (*p* adj ≤ 0.05, FDR ≤ 0.05, probability ≥ 0.95, and Log2 Fold Change (Log2FC) ≥ 2) were set as cut-off values. To annotate transcripts, TransDecoder software (version 5.5.0) was used (https://github.com/TransDecoder/TransDecoder.wiki.git accessed on 1 June 2022) to predict open reading frames (ORFs) and obtain protein sequences of at least 100 amino acids in length. Subsequently, transcripts with ORFs greater than 100 bp were aligned through Blastn and Blastx, limiting one match per transcript and ORF with a cut-off *e*-value of ≤10^−5^ against the SwissProt/UniProtKB database [79], and later, the HMMER v. 3.3.2 software [80] was used to compare the ORF against the PFAM database v. 33.1 [81,82,83].

### 4.5. Functional Classification by Gene Ontology (GO) and Assignment of Specific Metabolic Pathways

Blast2GO v. 6.0 software was used to visualize the transcripts´ functional annotations [84] with the gene ontology (GO) and enzyme code (EC) terms of the DEGs. GO terms were assigned from three main GO categories (molecular function, biological process, and cellular component). Likewise, the amino acid sequences of differentially expressed genes (DGEs) were selected for each condition evaluated (asymptomatic and symptomatic). The Kyoto Encyclopedia of Genes and Genomes (KEGG) database was used to perform the assignment of specific metabolic pathways. Additionally, KEGG BlastKOALA and GhostKOALA were used to automatically annotate and map sequences. The R-Studio package (https://www.rstudio.com accessed on 1 August 2022) was used to plot GO data as horizontal bar graphs and to perform a heatmap of DEGs expression data using the default options of package gplots. 

### 4.6. RNA Isolation and cDNA Synthesis

Total RNA was isolated from Persian lime (*Citrus latifolia* Tanaka) leaves from HLB-free (asymptomatic) and symptomatic trees naturally infected with *C*Las. The Quick-RNA MiniPrep Kit (Zymo Research) was used to isolate total RNA following the manufacturer´s instructions. Genomic DNA was extracted from samples using DNase I (Thermo Fisher Scientific, DNA-free kit, Ambion, Waltham, MA, USA). RNA concentration and purity were measured with a NanoDrop One^®^ (Thermo Scientific NanoDrop Technologies, LLC, Wilmington, DE, USA), and the quality was evaluated by 1.5% agarose gel electrophoresis during 30 min at 80 V. For double-stranded cDNA synthesis, 50 U/μL MultiScribe^TM^ Reverse Transcriptase was used, following the manufacturer’s protocol (Invitrogen/Life Technologies, Carlsbad, CA, USA). Reverse transcription conditions were 5 min at 25 °C, 60 min at 42 °C, and 15 min at 70 °C. 

### 4.7. Gene Expression Validation

We used real-time quantitative PCR (RT-qPCR) to validate the transcriptome sequencing data using a thermocycler CFX-96 Real-Time PCR System with 96-well PCR plates. (Bio-Rad, USA). Primer sequences were designed using Primer Express v. 3.0.1 Software (Applied Biosystems, Foster City, CA, USA). In addition, the primer pairs for eight candidate genes and the reference gene are shown in Appendix A. Efficiency for each gene was determined from a standard curve of serial dilutions of cDNA, and specificity was confirmed by a standard melting curve method. Amplification runs were initiated at 95 °C for 30 s, followed by 39 cycles of denaturation at 95 °C for 15 s, annealing at 59 °C for 30 s and extension at 72 °C for 30 s. The amplification system consisted of 10 μL of SsoAdvanced Universal Inhibitor-Tolerant SYBR^®^ Green Supermix for signal detection (Bio-Rad), 1 μL each of 5 μmol/L forward and reverse primers, 600 ng of cDNA template, and 7 μL of nuclease-free water for a total volume of 20 μL. Relative Expression Levels (REL) of target genes were calculated using the 2^−ΔΔCT^ method [85]. The relative quantification of the target genes was normalized using *ClUPL7* as the reference gene after its approval for stability by Bio-Rad CFX Maestro v 2.1 software. Expression analysis was carried out using three biological replicates and three technical replicates for each biological replicate. Expression analysis was subjected to a one-way analysis of variance (ANOVA) (*p* < 0.05) and Tukey’s test using Statgraphics Plus 5.1 software (Statistical Graphics Corp., The Plains, VA, USA). 

## 5. Conclusions

The present study suggests that Persian lime tolerance against HLB could be associated with a differential response in PTI activation through the *ClRPS2* gene and *ClHSP90*, compared with susceptible citrus genotypes. In addition, we identified some candidate genes within the starch and sucrose metabolism pathways that may be important to understand the carbohydrate metabolic response of Persian lime to the imbalance of starch accumulation. 

Additionally, the transcriptome and RT-qPCR analyses suggest that candidate genes associated with biotic stress are tentatively related to the tolerant response of Persian lime to *C*Las. The results of this research set the basis for further studies trying to understand the complex molecular interactions in the *C*Las-Persian lime pathosystem, and they may pave the way for genetic improvement of citrus aimed at the generation of HLB-tolerant cultivars.

## Figures and Tables

**Figure 1 ijms-24-07497-f001:**
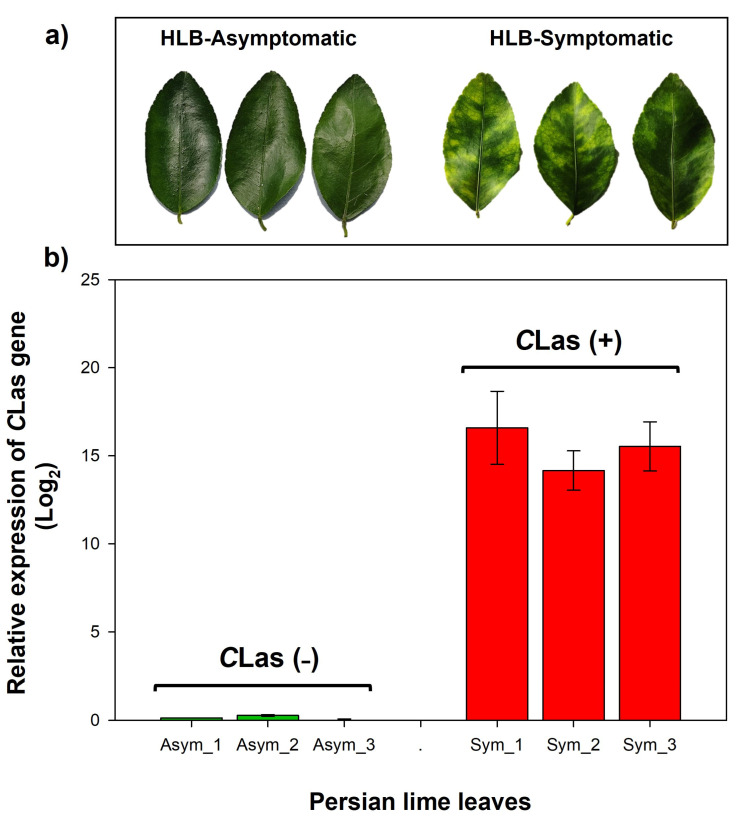
(**a**) HLB contrasting symptoms in Persian lime (*Citrus latifolia*) trees belonging to the Ixtacuaco Field Experiment (INIFAP) plot in Veracruz, Mexico. (**b**) RT-qPCR analysis of CLas detection in HLB-asymptomatic (Asym_1 to Asym_3) and HLB-symptomatic (Sym_1 to Sym_3) Persian lime (*C. latifolia*) leaves. Vertical bars represent means ± SD (*n* = 3 per group).

**Figure 2 ijms-24-07497-f002:**
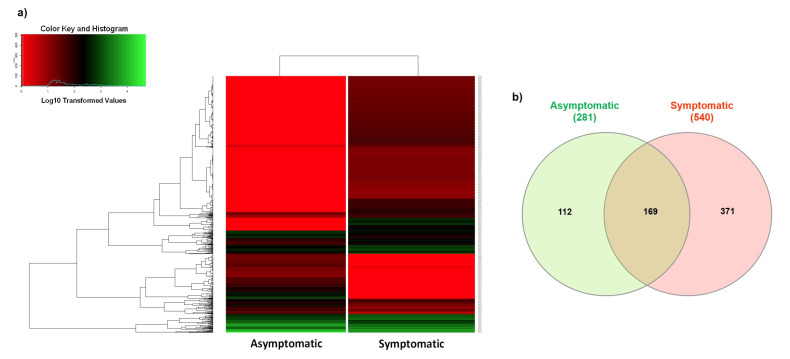
Analysis of DEGs in the Persian lime transcriptome. (**a**) Hierarchical clustering of 652 DEGs identified in the Persian lime transcriptome for two conditions (asymptomatic and symptomatic HLB) is shown. The heat map indicates downregulated transcripts in red, upregulated transcripts in green, and transcripts with similar expression patterns in black. (**b**) Venn diagram for DEGs in HLB-asymptomatic and HLB-symptomatic leaves. Numbers in the intersection indicate the number of detected genes with at least one read (gene label) in those intersections.

**Figure 3 ijms-24-07497-f003:**
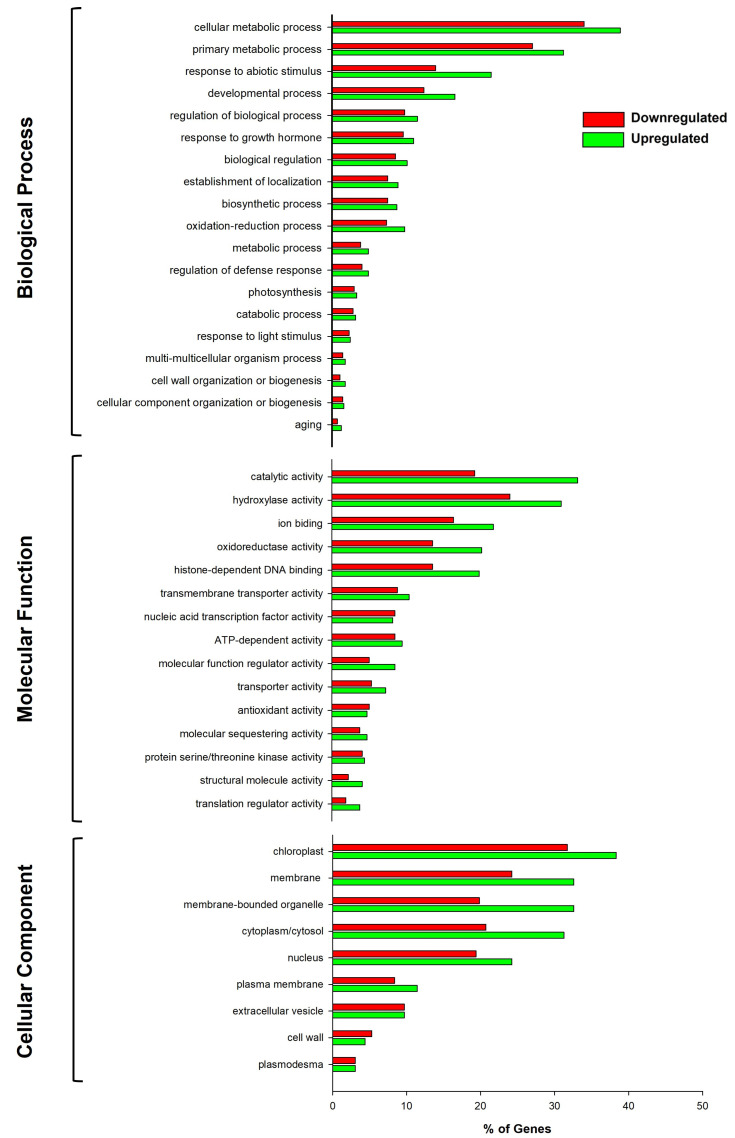
Gene ontology annotation analysis of DEGs in HLB-symptomatic Persian lime leaves. The percentage of genes is based on GO Slim terms for the three main categories of biological process, molecular function, and cellular component assigned to all genes.

**Figure 4 ijms-24-07497-f004:**
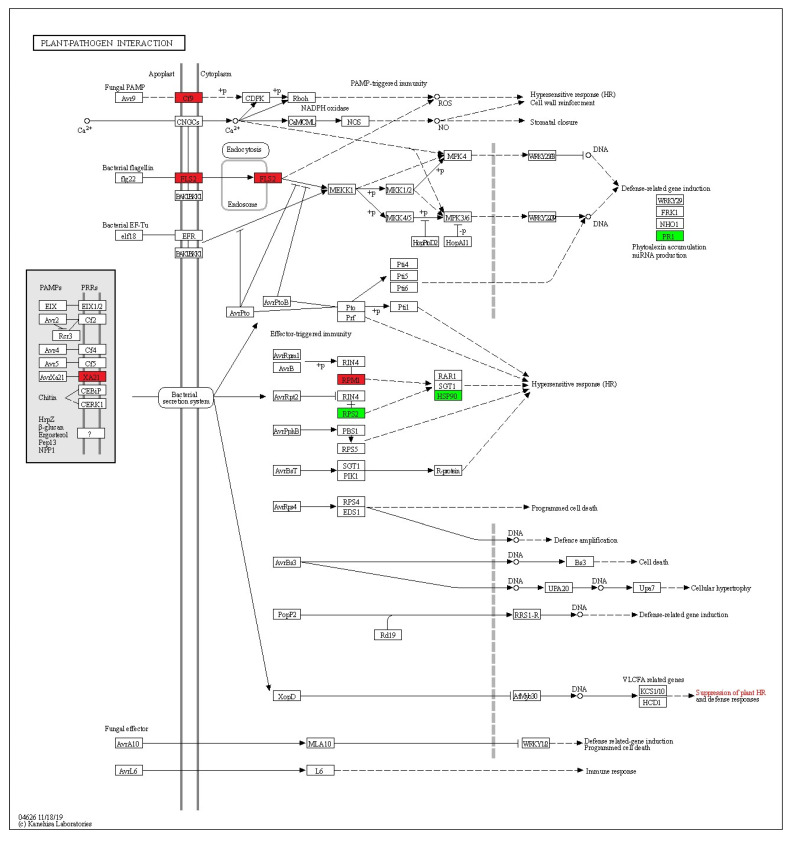
Transcripts related to plant–pathogen interactions in *C*Las-infected Persian lime leaves. The green boxes indicate upregulated genes, and red boxes indicate downregulated genes. The annotation was calculated with KEGG Blast Koala.

**Figure 5 ijms-24-07497-f005:**
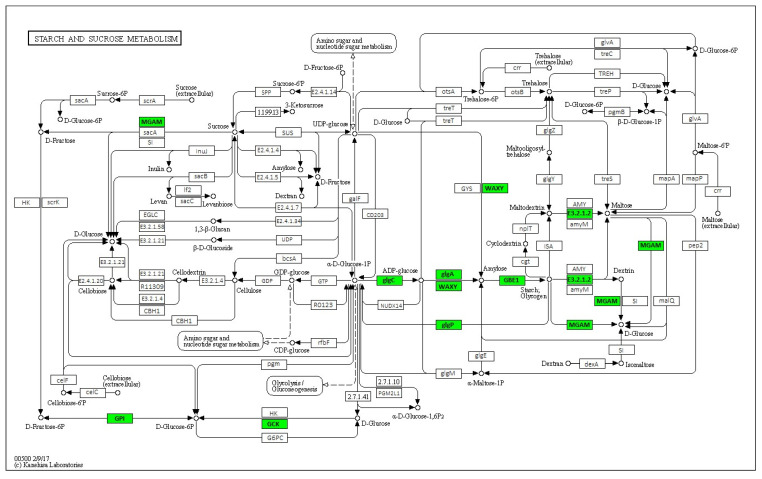
Transcripts related to starch and sucrose metabolism in *C*Las-infected Persian lime leaves. The green boxes indicate upregulated genes. The annotation was calculated with KEGG Blast Koala.

**Figure 6 ijms-24-07497-f006:**
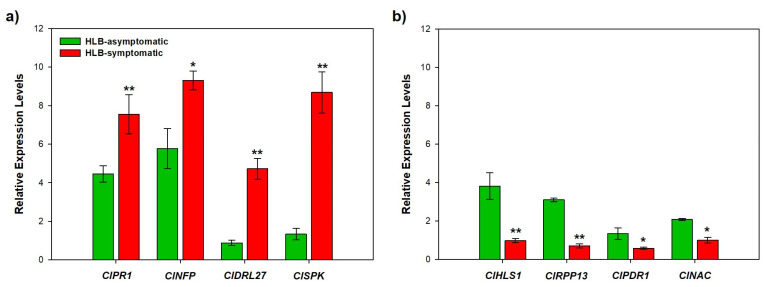
RT-qPCR validation of eight DEGs from the HLB diseased Persian lime transcriptome. Four upregulated (**a**) and four downregulated (**b**) DEGs by *C*Las infection in the transcriptome show similar Relative Expression Levels in *C*Las-infected (red bars) and healthy (green bars) Persian lime leaves. mRNA abundance was normalized using the housekeeping *UPL7* gene. Vertical bars represent the mean ± SD (*n = 3* for each group). Significant differences from controls are expressed as ** *p* < 0.01, * *p* < 0.05.

**Table 1 ijms-24-07497-t001:** Genes associated with signaling and disease resistance in response to HLB infection in the Persian lime transcriptome.

ID Gene	Name	BLASTP_Hit	Pfam	Expression Asym	Expression Sym	Fold Change (Log2)	Regulated
DN91145_c0_g1_i1	*ClNFP*	Serine/threonine receptor-like kinase NFP	Serine/threonine receptor-like kinase NFP	0	80.80	8.60	UP_Sym
DN14097_c0_g1_i1	*ClDRL27*	Disease resistance protein DRL27	Disease resistance protein	0	25.86	6.95	UP_Sym
DN6810_c0_g2_i1	*ClSRK*	G-type lectin S-receptor-like serine/threonine-protein kinase	G-type lectin S-receptor-like serine/threonine-protein kinase	0	20.20	6.60	UP_Sym
DN97321_c0_g1_i1	*ClCTR1*	Serine/threonine-protein kinase CTR1	Serine/threonine-protein kinase	0	16.85	6.34	UP_Sym
DN135982_c0_g1_i1	*ClRLP46*	Receptor-like protein 46 RLP46	Leucine Rich repeat	0	14.95	6.16	UP_Sym
DN20835_c0_g1_i1	*ClHSP90*	Heat shock protein 90-3 HSP	Hsp90 protein	0	14.53	6.12	UP_Sym
DN153599_c0_g1_i1	*ClRH56*	DEAD-box ATP-dependent RNA helicase 56 RH56	Helicase conserved C-terminal domain	0	12.89	5.95	UP_Sym
DN9557_c0_g1_i1	*ClPR1*	Basic form of pathogenesis-related protein 1 PR1	Cysteine-rich secretory protein family	5.39	111.16	4.35	UP_Sym
DN5017_c1_g1_i1	*ClWRKY23*	WRKY transcription factor 23	WRKY DNA-binding domain	12.57	54.84	2.11	UP_Sym

DN112451_c0_g1_i1	*ClHSL1*	Receptor-like protein kinase HSL1	Receptor-like protein	288.52	0	−10.80	DOWN_Sym
DN27097_c0_g1_i1	*ClRPPL1*	Putative disease resistance RPP13-like protein	Putative disease resistance RPP13-like protein 1	167.82	0	−10.02	DOWN_Sym
DN14613_c0_g1_i1	*ClPDR1*	Pleiotropic drug resistance protein 1	Pleiotropic drug resistance protein 1	161.81	0	−9.97	DOWN_Sym
DN92124_c0_g1_i1	*ClERF5*	Ethylene-responsive transcription factor 5	ERF domain transcription factor	87.10	0	−9.08	DOWN_Sym
DN8960_c0_g2_i1	*ClLAZ5*	Disease resistance protein LAZ5	Leucine Rich Repeat	78.15	0	−8.92	DOWN_Sym
DN180788_c0_g1_i1	*ClUCN*	Serine/threonine-protein kinase UCN	Protein tyrosine and serine/threonine kinase	68.64	0	−8.73	DOWN_Sym
DN19568_c0_g1_i2	*ClLRRC30*	Leucine Rich Repeat 30	Leucine Rich Repeat	61.24	0	−8.57	DOWN_Sym
DN13414_c0_g1_i2	*ClRLP54*	Receptor-like protein 54 RLP54	Leucine Rich Repeat	37.33	0	−7.86	DOWN_Sym
DN10515_c0_g1_i1	*ClNPR1*	NPR1_interact	NPR1 interacting	132.26	17.86	−3.22	DOWN_Sym
DN18490_c0_g1_i1	*ClLRK71*	L-type lectin-domain containing receptor kinase VII.1 LRK71	Legume lectin domain	24.23	4.10	−2.56	DOWN_Sym
DN18226_c0_g1_i2	*ClSSL10*	Protein Strictosidine synthase-like 10	Strictosidine synthase	97.02	21.34	−2.19	DOWN_Sym
DN10036_c0_g2_i1	*ClNAC73*	NAC domain-containing protein 73 NAC73	No apical meristem (NAM) protein	101.89	23.28	−2.13	DOWN_Sym

## Data Availability

The data generated are available upon a reasonable requisition.

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
