# Peer review of "Insights into the Molecular Basis of Huanglongbing Tolerance in Persian Lime (Citrus latifolia Tan.) through a Transcriptomic Approach"

_ijms, 2023, doi:10.3390/ijms24087497_

Round 1

Reviewer 1 Report

The reviewer found the work to be important and interesting, but the overall manuscript read like the authors are not interpreting and presenting RNA-Seq results in proper ways. This reviewer is addressing those issues for the authors to make corrections and resubmit the manuscript after thorough revision.

The authors identified differentially expressed (DE) genes from contrasts between symptomatic (infected) vs asymptomatic (control) leaves. The presentation of results should be made in reference to gene expressions in the symptomatic leaf. For example, up-regulated or down-regulated in symptomatic leaves only, because that tells the readers about the treatment effect (i.e., infected vs uninfected control). Up-regulation or down-regulation in asymptomatic or control comes from comparing their transcripts levels to the symptomatic ones; so, no need to say if gene(s) shows differential expression in asymptomatic. Figure 2b shows this redundancy where the set of up-regulated genes in asymptomatic is same set that is down-regulated in symptomatic and vice versa. The manuscript needs to avoid this redundancy throughout the manuscript. However, the authors do not seem to comprehend the fact that differential expression is a contrast. Case in point, they mention “Interestingly, these same genes found in starch and sucrose metabolism were found to be downregulated in asymptomatic-HLB leaves (Lines 181-182).” Of course, they would find that since significant DE genes upregulated in one condition means that it is downregulated in the other; because that is the contrast that DESeq2 tested and found significant. On the other hand, the FPKM/TPM values (Figure 2a, perhaps?) can be turned into heatmaps with both symptomatic and asymptomatic samples displaying gene expression levels independent of each other.

Other comments:

A few questions in reference to the following sentence in your abstract: “Also, the KEGG analysis suggests that the tolerance against HLB could be mediated by ClRSP2 and ClHSP90 genes, unlike the observed in susceptible citrus genotypes.” The reviewer did not see authors using multiple genotypes, both resistant and susceptible, being tested. What is the basis of suggesting that the two genes do not mediate tolerance against HLB in the susceptible citrus genotypes? How can KEGG analysis suggest whether or not the genes mediate tolerance against HLB?

Line 55: Please also use the common name, citrus psyllid.

Lines 61-62: “the cellular integrity of the plant it’s compromised due to Reactive Oxygen Species”. Maybe “is” NOT “it’s”?

Line 69: “species” NOT “specie” (?)

Comments In reference to the following sentence (Lines 73-75): To understand the plant response to the CLas infection, multiple transcriptomic analyzes of gene expression profiles by RNA sequencing (RNA-Seq) were performed with the aim of better understanding host–pathogen interactions.

“To understand” and “with the aim of better understanding” are redundant, please rephrase. “analyses” NOT “analyzes”.

Line 75-76: You mention “studies” but only produce one reference (?)

Lines 77-86: reference 23 needs to be moved to first mention of the study.

Comments about Figure 2: The image resolution does not allow one to read some very small fonts even at higher zoom levels. Picture quality needs to be upgraded and fonts need to be increased. Figure 2a title “Citrus x latifolia in response to HLP” does not make any sense.

Lines 149-150: “notorious” is not a word for this context.

Comment about Figure 3: GO annotation is not based on DEGs because DEGs would be in contrasts between symptomatic vs asymptomatic samples and the figure would only show up- or downregulated categories for one of the conditions compared to the other condition.

Figure 6: the charts do not show “Relative Expression Levels”. It shows expression levels for the genes under two different conditions. The controls against which the significant differences were drawn were HLB-asymptomatic. The asterisks shown in the qPCR charts is analogous to LOD of 2 for DEGs. That means, DEGs are declared when expression contrasts between HLB-symptomatic and HLB-asymptomatic are significant. Going back to the main limitation of this manuscript with RNA-Seq interpretation and presentation, the authors seem to use this contrast to explain identified DEGs as they were independent between the two conditions. 

This reviewer would like to see a thoroughly revised manuscript.

Author Response

Reviewer 1

The reviewer found the work to be important and interesting, but the overall manuscript read like the authors are not interpreting and presenting RNA-Seq results in proper ways. This reviewer is addressing those issues for the authors to make corrections and resubmit the manuscript after thorough revision.

The authors identified differentially expressed (DE) genes from contrasts between symptomatic (infected) vs asymptomatic (control) leaves. The presentation of results should be made in reference to gene expressions in the symptomatic leaf. For example, up-regulated or down-regulated in symptomatic leaves only, because that tells the readers about the treatment effect (i.e., infected vs uninfected control). Up-regulation or down-regulation in asymptomatic or control comes from comparing their transcripts levels to the symptomatic ones; so, no need to say if gene(s) shows differential expression in asymptomatic. Figure 2b shows this redundancy where the set of up-regulated genes in asymptomatic is same set that is down-regulated in symptomatic and vice versa. The manuscript needs to avoid this redundancy throughout the manuscript. However, the authors do not seem to comprehend the fact that differential expression is a contrast. Case in point, they mention “Interestingly, these same genes found in starch and sucrose metabolism were found to be downregulated in asymptomatic-HLB leaves (Lines 181-182).” Of course, they would find that since significant DE genes upregulated in one condition means that it is downregulated in the other; because that is the contrast that DESeq2 tested and found significant. On the other hand, the FPKM/TPM values (Figure 2a, perhaps?) can be turned into heatmaps with both symptomatic and asymptomatic samples displaying gene expression levels independent of each other.

This comment was very important for the improvement of the manuscript. The current version of the manuscript consider only the upregulation or downregulation of gene expression in the HLB-symptomatic leaves compared with the healthy non-symptomatic leaves. This comment was applied in all manuscript. 

Other comments:

A few questions in reference to the following sentence in your abstract: “Also, the KEGG analysis suggests that the tolerance against HLB could be mediated by ClRSP2 and ClHSP90 genes, unlike the observed in susceptible citrus genotypes.” The reviewer did not see authors using multiple genotypes, both resistant and susceptible, being tested. What is the basis of suggesting that the two genes do not mediate tolerance against HLB in the susceptible citrus genotypes? How can KEGG analysis suggest whether or not the genes mediate tolerance against HLB?

The abstract was rewritten. The downregulated activity of the RSP2 and HSP90 genes in susceptible genotypes such as C. sinensis has been observed in previous studies, in contrast to the observed in our study of the tolerant C. latifolia.

Line 55: Please also use the common name, citrus psyllid.

Attended.

Lines 61-62: “the cellular integrity of the plant it’s compromised due to Reactive Oxygen Species”. Maybe “is” NOT “it’s”?

Attended.

Line 69: “species” NOT “specie” (?)

Attended.

Comments In reference to the following sentence (Lines 73-75): To understand the plant response to the CLas infection, multiple transcriptomic analyzes of gene expression profiles by RNA sequencing (RNA-Seq) were performed with the aim of better understanding host–pathogen interactions.

“To understand” and “with the aim of better understanding” are redundant, please rephrase. “analyses” NOT “analyzes”.

Attended.

Line 75-76: You mention “studies” but only produce one reference (?)

 More references were included in this phrase.

Lines 77-86: reference 23 needs to be moved to first mention of the study.

Attended.

Comments about Figure 2: The image resolution does not allow one to read some very small fonts even at higher zoom levels. Picture quality needs to be upgraded and fonts need to be increased. Figure 2a title “Citrus x latifolia in response to HLP” does not make any sense.

The quality of Figure 2 was improved and the tittle was removed. Also the graph with the up and downregulated genes in symptomatic and asymptomatic leaves was removed to be in line with the first comment of the reviewer.

Lines 149-150: “notorious” is not a word for this context.

Attended.

Comment about Figure 3: GO annotation is not based on DEGs because DEGs would be in contrasts between symptomatic vs asymptomatic samples and the figure would only show up- or downregulated categories for one of the conditions compared to the other condition.

We modified Figure 3 to consider the number of genes in each GO category that is up or down-regulated in the HLB-symptomatic leaves, instead of plotting symptomatic vs asymptomatic.

Figure 6: the charts do not show “Relative Expression Levels”. It shows expression levels for the genes under two different conditions. The controls against which the significant differences were drawn were HLB-asymptomatic. The asterisks shown in the qPCR charts is analogous to LOD of 2 for DEGs. That means, DEGs are declared when expression contrasts between HLB-symptomatic and HLB-asymptomatic are significant. Going back to the main limitation of this manuscript with RNA-Seq interpretation and presentation, the authors seem to use this contrast to explain identified DEGs as they were independent between the two conditions. 

This reviewer would like to see a thoroughly revised manuscript.

The Figure 6 is the validation of the Transcriptomic data through RT-qPCR. We use the term “Relative Expression Level” (REL) considered as the expression level of a gene of interest against an expression of an internal control gene. The relativity of the expression on this graph is not considered among the symptomatic and asymptomatic leaves, in fact, we are showing the REL of the gene in the two conditions (symptomatic and asymptomatic). REL is the conventional way to express RT-qPCR results.

Reviewer 2 Report

The authors report different gene expression between symptomatic and non-symptomatic trees to citrus greening disease in Persian lime. The results will be interesting for researchers on this disease or plant genetics as well as plant physiologists. This manuscript will be published in the journal with moderate modification. Some notices on the manuscript are written below.

1. English

English should be checked by a native English speaker and the authors have to correct typographical and grammatical errors in the manuscript before the resubmission of a revised manuscript.

L17-18

Las is not the only pathogen attributed to citrus greening disease. The sentence should be changed.

L55

Do not abbreviate the author of the insect.

L94

When the scientific name of the species studied is referred to first, spell out both the genus and species with the author of the species.

Figure 1

Three photos of symptomatic leaves, especially the right one, are so brilliant that they appear symptomatic. Replace them with not brilliant(mat) photos. The authors of the species do not need to be referred to here but italicize the scientific name. Explain what bars indicate. The statistical results wrongly marked. Since ND and asterisks are shown over the green and red bars, respectively, this means that the means were not significantly different among symptomatic trees but were significantly different among symptomatic trees. Check the results.

L116

Explain how the probability was adjust and why. Is it sure for “”? Which sign, equality or inequality is used depends on the null hypothesis. Check it.

Figure 2

Letters in the graphs are too small to read. Change the font size.

L148

It seems better to write “stimuli”, not stimulus, in this context.

Figure 5

Green should be changed for another colour, since the gene expressed green function different those of greened genes in Figure 4.

L195-196

Either “≥” or “≤” should be replaced for “>” or “<”, otherwise 2 is included in bother categories.

L217-227

Move these sentences to Introduction.

L241

Probably it should be “species”, not “specie”.

L344

The author of the species should be removed (see above for the reference to the author of the species).

After L344

The authors should elaborate on their collection of materials and date: how many leaves or trees were collected when. What size were the plants in. This rule should be applied throughout this manuscript.

L358

“G”, not “g”?

L363

“n”, not “η”?

Finally one issue is pointed out: What biological or agricultural differences are seen between symptomatic and non-symptomatic? The statement of the comparison will facilitate the understanding of readers.

Author Response

Reviewer 2

The authors report different gene expression between symptomatic and non-symptomatic trees to citrus greening disease in Persian lime. The results will be interesting for researchers on this disease or plant genetics as well as plant physiologists. This manuscript will be published in the journal with moderate modification. Some notices on the manuscript are written below.

  1. English

English should be checked by a native English speaker and the authors have to correct typographical and grammatical errors in the manuscript before the resubmission of a revised manuscript

An expert has checked the English language correctness. All typographical and grammatical errors have now been corrected in the revised version of the MS.

L17-18

Las is not the only pathogen attributed to citrus greening disease. The sentence should be changed.

The sentence was modified to mention the others Ca. Liberibacter species that cause HLB in other parts of the world.

L55

Do not abbreviate the author of the insect.

Attended.

 L94

When the scientific name of the species studied is referred to first, spell out both the genus and species with the author of the species.

Attended.

 Figure 1

Three photos of symptomatic leaves, especially the right one, are so brilliant that they appear symptomatic. Replace them with not brilliant(mat) photos. The authors of the species do not need to be referred to here but italicize the scientific name. Explain what bars indicate. The statistical results wrongly marked. Since ND and asterisks are shown over the green and red bars, respectively, this means that the means were not significantly different among symptomatic trees but were significantly different among symptomatic trees. Check the results.

Figure 1 was modified to attend this comment.

L116

Explain how the probability was adjust and why. Is it sure for “≥”? Which sign, equality or inequality is used depends on the null hypothesis. Check it.

In transcriptomic analysis, the values p and p-adjusted (also named q-value) are used to determine the genes with differential expression (DEGs). For the adjusted p-value, the False Discovery Rate (FDR) is applied to accept that 5% of the identified DEGs in the transcriptomic data may be false positives. However, we consider that this explanation is not necessary in the manuscript because the use of the p-adjusted value is a widely accepted practice in genomic sciences. The use of > and < was checked.

Figure 2

Letters in the graphs are too small to read. Change the font size.

 Attended.

L148

It seems better to write “stimuli”, not stimulus, in this context.

 Attended.

Figure 5

Green should be changed for another colour, since the gene expressed green function different those of greened genes in Figure 4.

 The Figures 4 and 5 represent different metabolic routes, the green or red color in the genes represent up or downregulation in each route, that is why we consider that it is not necessary to change the color in Figure 5.

L195-196

Either “≥” or “≤” should be replaced for “>” or “<”, otherwise 2 is included in bother categories.

 Attended.

L217-227

Move these sentences to Introduction.

 We consider that this paragraph is a proper introduction of the discussion, as the ideas discussed  in this paragraph, have been already included in the introduction.

L241

Probably it should be “species”, not “specie”.

 Attended.

L344

The author of the species should be removed (see above for the reference to the author of the species).

Attended.

 After L344

The authors should elaborate on their collection of materials and date: how many leaves or trees were collected when. What size were the plants in. This rule should be applied throughout this manuscript.

 Attended.

L358

“G”, not “g”?

 Attended.

L363

“n”, not “η”?

 Attended.

Finally, one issue is pointed out: What biological or agricultural differences are seen between symptomatic and non-symptomatic? The statement of the comparison will facilitate the understanding of readers.

The visual differences between symptomatic and asymptomatic leaves are those shown in Figure 1.

Round 2

Reviewer 1 Report

This reviewer would like to thank the authors for their diligent work on his earlier comments/suggestions. Please find some additional comments/suggestions that you may or may not agree with, but this reviewer would like to point them out so that you may improve your manuscript. 

ABSTRACT: “that may set the basis to develop strategies for the integrated management of this important Citrus disease” HOW? The discussion does not talk about these strategies.

INTRODUCTION:

Line 52: Citrus crops are not immune to diseases” Is it needed?

Lines 86-92: The sentence talks about the results of studies investigating HLB-tolerant citrus genotypes. It cannot start with “For instance”. Maybe, “One such study showed” (?) Also, ‘few studies” on HLB-tolerant also need to be cited. There is just one [23] being acknowledged. Instead of a snapshot of one of the studies, can you summarize those “few studies” (?)

Lines 98-102: What were the gaps in the earlier studies that you would want to fill? Maybe, start the last paragraph with those gaps to inform readers that you are not simply adding one more research, but you are trying to fill the gap. There is a sentence in the “Discussion” that reads “Although several studies have reported DEGs in HLP………, the identification and expression of DEGs in Persian lime, a species considered HLB-tolerant, had not been studied.” Maybe, this sentence needs to be moved to introduction. You may highlight this point later in the discussion section too.

RESULTS:

Lines 125-128: The last sentence is not needed. You can simply refer to “Supplementary File 1” at the end of the previous sentence.

Figures 4 and 5: Perhaps figures come from an online pathway tool. The authors can come up with innovative ways to reduce clutter by removing unnecessary parts of those pathways. Need to increase font size to make it legible, manual editing, maybe (?) This reviewer will defer this judgement to the editor.

DISCUSSION:

Line 264: “This approach may be especially relevant” does not read well. Earlier statement is not an approach.

General observation: highly expressed vs. upregulated should be used consistently. It is customary to adhere to use consistent terminology throughout the manuscript.

Author Response

ABSTRACT: “that may set the basis to develop strategies for the integrated management of this important Citrus disease” HOW? The discussion does not talk about these strategies.

INTRODUCTION:

Line 52: Citrus crops are not immune to diseases” Is it needed?

Lines 86-92: The sentence talks about the results of studies investigating HLB-tolerant citrus genotypes. It cannot start with “For instance”. Maybe, “One such study showed” (?) Also, ‘few studies” on HLB-tolerant also need to be cited. There is just one [23] being acknowledged. Instead of a snapshot of one of the studies, can you summarize those “few studies” (?)

Lines 98-102: What were the gaps in the earlier studies that you would want to fill? Maybe, start the last paragraph with those gaps to inform readers that you are not simply adding one more research, but you are trying to fill the gap. There is a sentence in the “Discussion” that reads “Although several studies have reported DEGs in HLP………, the identification and expression of DEGs in Persian lime, a species considered HLB-tolerant, had not been studied.” Maybe, this sentence needs to be moved to introduction. You may highlight this point later in the discussion section too.

RESULTS:

Lines 125-128: The last sentence is not needed. You can simply refer to “Supplementary File 1” at the end of the previous sentence.

Figures 4 and 5: Perhaps figures come from an online pathway tool. The authors can come up with innovative ways to reduce clutter by removing unnecessary parts of those pathways. Need to increase font size to make it legible, manual editing, maybe (?) This reviewer will defer this judgement to the editor.

DISCUSSION:

Line 264: “This approach may be especially relevant” does not read well. Earlier statement is not an approach.

General observation: highly expressed vs. upregulated should be used consistently. It is customary to adhere to use consistent terminology throughout the manuscript.

All comments were attended in the submitted manuscript. 
